# Simultaneous Multi-Organ Metastases from Chemo-Resistant Triple-Negative Breast Cancer Are Prevented by Interfering with WNT-Signaling

**DOI:** 10.3390/cancers11122039

**Published:** 2019-12-17

**Authors:** Iram Fatima, Ikbale El-Ayachi, Hilaire C. Playa, Jackelyn A. Alva-Ornelas, Aysha B. Khalid, William L. Kuenzinger, Peter Wend, Jackelyn C. Pence, Lauren Brakefield, Raisa I. Krutilina, Daniel L. Johnson, Ruth M. O’Regan, Victoria Seewaldt, Tiffany N. Seagroves, Susan A. Krum, Gustavo A. Miranda-Carboni

**Affiliations:** 1Department of Medicine, College of Medicine at UTHSC (University of Tennessee Health Science Center), UTHSC Center for Cancer Research Memphis, Memphis, TN 38163, USA; iram.fatima@unmc.edu (I.F.); Ikbale.ElAyachi@osumc.edu (I.E.-A.); willkuen@uthsc.edu (W.L.K.); lbrakefi@uthsc.edu (L.B.); 2Department of Pathology and Laboratory Medicine, College of Medicine at UTHSC, UTHSC Center for Cancer Research, Memphis, TN 38163, USA; hplaya@uthsc.edu (H.C.P.); rkrutili@uthsc.edu (R.I.K.); tseagro1@uthsc.edu (T.N.S.); 3Department of Population Science, City of Hope Comprehensive Cancer Center and Beckman Institute, Duarte, CA 91010, USA; jalvao@coh.org (J.A.A.-O.); vseewaldt@coh.org (V.S.); 4Department of Orthopaedic Surgery and Biomedical Engineering, UTHSC Center for Cancer Research, UTHSC, Memphis, TN 38163, USA; khalidaysha2000@gmail.com (A.B.K.); jpence3@uthsc.edu (J.C.P.); smirand5@uthsc.edu (S.A.K.); 5Department of Obstetrics and Gynecology, David Geffen School of Medicine at UCLA, Jonsson Comprehensive Cancer Center, Los Angeles, CA 90095, USA; peterwend@web.de; 6Molecular Bioinformatics Core, UTHSC, Memphis, TN 38163, USA; djohn166@uthsc.edu; 7Departments of Medicine, Division of Hematology and Oncology, University of Wisconsin School of Medicine and Public Health, Madison, WI 53705, USA; roregan@medicine.wisc.edu

**Keywords:** triple-negative breast cancer, metastasis, inhibiting WNT/β-catenin signaling

## Abstract

Triple-negative breast cancers (TNBCs), which lack specific targeted therapy options, evolve into highly chemo-resistant tumors that metastasize to multiple organs simultaneously. We have previously shown that TNBCs maintain an activated WNT10B-driven network that drives metastasis. Pharmacologic inhibition by ICG-001 decreases β-catenin-mediated proliferation of multiple TNBC cell lines and TNBC patient-derived xenograft (PDX)-derived cell lines. In vitro, ICG-001 was effective in combination with the conventional cytotoxic chemotherapeutics, cisplatin and doxorubicin, to decrease the proliferation of MDA-MB-231 cells. In contrast, in TNBC PDX-derived cells doxorubicin plus ICG-001 was synergistic, while pairing with cisplatin was not as effective. Mechanistically, cytotoxicity induced by doxorubicin, but not cisplatin, with ICG-001 was associated with increased cleavage of PARP-1 in the PDX cells only. In vivo, MDA-MB-231 and TNBC PDX orthotopic primary tumors initiated de novo simultaneous multi-organ metastases, including bone metastases. WNT monotherapy blocked multi-organ metastases as measured by luciferase imaging and histology. The loss of expression of the WNT10B/β-catenin direct targets HMGA2, EZH2, AXIN2, MYC, PCNA, CCND1, transcriptionally active β-catenin, SNAIL and vimentin both in vitro and in vivo in the primary tumors mechanistically explains loss of multi-organ metastases. WNT monotherapy induced VEGFA expression in both tumor model systems, whereas increased CD31 was observed only in the MDA-MB-231 tumors. Moreover, WNT-inhibition sensitized the anticancer response of the TNBC PDX model to doxorubicin, preventing simultaneous metastases to the liver and ovaries, as well as to bone. Our data demonstrate that WNT-inhibition sensitizes TNBC to anthracyclines and treats multi-organ metastases of TNBC.

## 1. Introduction

Triple-negative breast cancers (TNBC) are an aggressive breast cancer subtype devoid of estrogen receptor (ER), progesterone receptor (PR) and human epidermal growth factor receptor 2 (HER2) amplification, and are, therefore, unresponsive to targeted therapies such as trastuzumab and anti-estrogen therapies. Many TNBC exhibit de novo resistance to chemotherapy and all metastatic TNBC eventually develop resistance [1]. TNBC disproportionately affects BRCA1 mutation carriers and young African-American women. There is evidence that African-American (AA) women diagnosed with TNBC have worse clinical outcomes than women of European-American (EA) descent [2]. The development of novel or repurposed drug therapies is urgently needed to treat highly chemo-resistant metastatic TNBC disease (metTNBC).

Metastatic breast cancer (MBC), also called stage IV or advanced breast cancer, is the leading cause of worldwide cancer-related deaths among women. ER+ cancers preferentially disseminate to the bone, liver, and brain [3]. In contrast, metTNBC preferentially metastasizes to multiple visceral organs, including the liver and lungs, to the brain and, less frequently, to the bone. metTNBC patients diagnosed with multi-organ metastases do worse and their median survival is about a year [4]. Patients identified by the WNT/β-catenin network classifier have a greater risk of lung and brain metastases in TNBC [5].

There are 19 known Wnt ligands that control a vast number of biological phenomena during fetal and post-natal development throughout adult life [6]. The canonical WNT pathway activates co-receptors (LRP5/6/FZD), subsequently leading to the stabilization of β-catenin, preventing the destruction complex, AXIN1/APC/GSK3β, from degrading β-catenin. In contrast, other WNT ligands are independent of β-catenin signaling. Aberrant WNT signaling underlies a wide range of pathologies in human disease, including some initiated by Wnt10B signaling [7,8]. We have previously shown that WNT10B expression (but not WNT1) is elevated in the majority of metTNBC and is predictive of poor outcome. Furthermore, the expression of the WNT10B downstream target HMGA2 alone correlates with metastasis [9]. Recently, our group established that a WNT10B-network, composed of β-catenin/HMGA2/EZH2, is associated with survival and metastases in chemo-resistant TNBC [10]. We now provide evidence that the WNT10B-network that characterizes aggressive metTNBC disease is conserved in four TNBC subtypes: mesenchymal (MSL), basal-like 1 (BSL1) [11], and in two TNBC patient-derived xenograft (PDX)-derived metastatic cell lines (cHCI-2 and cHCI-10), and that all models are responsive to WNT inhibition.

Herein, we demonstrate that the WNT inhibitor ICG-001, as a monotherapy, prevents simultaneous multi-organ and bone metastases in vivo in a traditional TNBC cell line model (MDA-MB-231; MSL subtype) and in a doxorubicin-resistant metTNBC PDX tumor model (HCI-10Luc2). Furthermore, we have shown that ICG-001 is synergistic with doxorubicin in preventing lung metastasis [10], whereas in this study, we now show that doxorubicin monotherapy does not repress multi-organ metastases. In contrast, when ICG-001 is combined with doxorubicin in the chemo-resistant HCI-10 TNBC PDX model, we can block both liver and bone metastases simultaneously.

Overall, our results demonstrate that WNT inhibition is effective in combination with doxorubicin and could provide a new therapeutic approach for chemo-resistant metTNBC.

## 2. Results

### 2.1. The WNT Inhibitor ICG-001 Inhibits the Growth and β-Catenin Gene Transcription of TNBC Cells

We previously demonstrated that WNT10B has epistatic activity on HMGA2 in MDA-MB-231 cells, regulating WNT10B-mediated proliferation [9]. Moreover, we showed that ICG-001 targets an auto-regulatory-loop, composed of HMGA2 and EZH2, that is necessary for maintenance of the WNT nuclear components β-catenin/TCF4/LEF-1 to induce gene transcription of WNT10B direct target genes [10]. To determine the effects of ICG-001 on proliferation of a subset of TNBC cell lines, we directly compared cell lines from two different TNBC subtypes, mesenchymal stem like (MSL: MDA-MB-231 and MDA-MB-157) and basal-like 1 (BL1: HCC-38 and MDA-MB-468). We also included cells derived from two TNBC PDX tumor models, one chemo-naïve (cHCI-2, never exposed to therapy) and one highly resistant to doxorubicin (cHCI-10) Figure 1A. Utilizing WST-1 proliferation assays, we determined the IC_50_ dose for ICG-001 following 48 h of exposure to increasing dosages ranging from 0.02 µM to 30 µM (0.02, 0.04, 0.2, 1, 5, 10, 20 and 30 µM). The ICG-001 effects are specific for TNBC, as exposure of ICG-001 was nontoxic to normal human mammary epithelial cells (HUMEC), and human epithelial breast MCF10A cells, as well as ER+ MCF7 and HER2+ SKBr3 cell lines (Appendix A). The cHCI-2 chemo-naïve cells were the most sensitive to ICG-001, whereas the chemo-resistant cHCI-10 cells were the most resistant. The mean IC_50_ levels for all cell lines are shown in Appendix A; all downstream experiments were conducted with the IC_50_s shown in Appendix A.

Next, to determine the effects of ICG-001 on known WNT10B/β-catenin direct target genes (*AXIN2* and *HMGA2)* [9,10] and proliferation-associated genes (*CCND1, MYC*, and *PCNA),* we treated the cells at their respective IC_50_ concentrations for 48 h as follows: MDA-MB-231 (10 μM) and MDA-MB-157 (20 μM, Figure 1BI,II) or cHCI-2 (6 μM) and cHCI-10 (35 μM, Figure 1CI,II) cells and then conducted qPCR analysis (Appendix A). There was a significant downregulation of *HMGA2* mRNA in MDA-MB-231, cHCI-2 and cHCI-10 cells (* *p <* 0.05 or ** *p <* 0.01). In contrast, the MDA-MB-157 cells did not show a significant change. Interestingly, MDA-MB-231 cells and cHCI-10 cells had a similar significant downregulation of mRNA expression for *CCND1, MYC* and *PCNA* (* *p* = 0.05 to ** *p* = 0.01) and these results were confirmed by immunoblotting (Figure 1D,E). In MDA-MB-157 cells, the mRNA expression was reduced significantly only for *CCND1*, *MYC* and *PCNA* (** p* = 0.01) and this was confirmed by immunoblotting.

The above results suggest that ICG-001 repressed WNT direct target genes by the disruption of transcriptionally active β-catenin, also known as ABC (i.e., lacks phosphorylation at amino acids Ser33/Ser37/Thr41) that is co-localized in the nucleus [12]. To test for that possibility, we performed immunoblotting for ABC in MDA-MB-231, MDA-MB-157, cHCI-2 and cHCI-10 cells at the appropriate IC_50_ dose for 48 h (Figure 1F). In response to ICG-001, ABC protein expression is decreased in the TNBC cell lines. Pan-β-catenin and β-actin served as controls. Immunoblots were quantified in a set of biological triplicates that demonstrated statistically significant changes of the immunoblots when treated with ICG-001 relative to the controls for each cell line tested (Appendix A). Taken together, the data suggest that ICG-001 decreases proliferation and reduces the expression of WNT10B/β-catenin direct target genes in a variety of TNBC cell lines.

### 2.2. The WNT Inhibitor ICG-001 Preferentially Synergizes with Doxorubicin, But Not Cisplatin, in Highly Chemo-Resistant TNBC Cells

It is known that β-catenin contributes to resistance to doxorubicin and to cisplatin in MDA-MB-231 cells, as the silencing of β-catenin expression restores drug sensitivity [13]. We have shown that, in an TNBC cell line (cHCI-10), ICG-001 is capable of sensitizing cells to doxorubicin and that this effect was synergistic by isobologram and combination index analysis [10]. However, the ability of ICG-001 to sensitize these cells to another FDA-approved frontline chemotherapy drug against TNBC, such as cisplatin (CIS), is unknown.

To test for synergy of ICG-001 with cisplatin, we next used sub-IC_50_ concentrations of ICG-001, either at 1 µM or 5 µM, for MDA-MB-231 cells, or at 5 µM and 10 µM for cHCI-10 cells in combination with various CIS dosages ranging from 0.02 μM to 20 μM (0.02, 0.04, 0.2, 1, 5, 10, and 20 μM) (Figure 2). WST-1 proliferation assays were performed every 24 h up to 96 h (Figure 2Ai or Figure 2Bi) and 48 h is shown in Appendix A. In MDA-MB-231 cells, ICG-001 (5 μM) was capable of reaching the IC_50_ threshold with as little as 0.04 μM of CIS by 96 h. As a monotherapy, CIS required 125-fold (5 μM) more drug to reach the IC_50_ threshold by 96 h. In contrast, ICG-001 was not able to reach the IC_50_ threshold in combination with CIS in the cHCI-10 cells until the ranges reached 20 μM by 96 h. Similar effects were observed in the cHCI-10 cells at 48 h (Appendix A). The patient from which the HCI-10 model was derived was not treated with a platinum agent, but died after treatment with cyclophosphamide, anthracyclines and taxanes [14]. Of note, there were no effects on proliferation with the drug combination at 24 h in either the MDA-MB-231 or cHCI-10 cell lines. The above data suggest that ICG-001 was more effective at sensitizing CIS-based therapy in the MDA-MB-231 cells than in the PDX-derived cells. We determined IC_30_, IC_50_ and IC_70_ values and plotted isobole curves to determine the combination index, but we found no additive or synergistic effects between cisplatin with ICG-001 [10,15] as the combination indexes were greater than 1.

Next, we repeated the above experiment with the same sub-IC_50_ concentrations of ICG-001 with increasing DOX dosages ranging from 0.02 to 20 μM (0.02, 0.04, 0.2, 1, 2, 10 and 20 μM) for both MDA-MB-231 and cHCI-10 cells (Figure 2A–D). In the MDA-MB-231 cells, the sub-IC_50_ concentrations of ICG-001 (1 μM and/or 5 μM) were capable of sensitizing cells to DOX therapy at 0.04 μM by 96 h (Figure 2Aii). In contrast, the DOX alone treatment required 25-fold higher concentrations (1 μM) to reach this threshold. There were no effects on proliferation at 24 h. The 48-h time point is shown for the MDA-MB-231 cells in Appendix A, demonstrating similar trends as the 96-h analysis.

The cHCI-10 cells were found to be more sensitive to the combinatorial therapies of ICG-001 plus DOX reaching the sub-IC_50_ threshold (~48–49%) (Figure 2). Importantly, at 96 h, ICG-001 was able to sensitize DOX with the three lowest concentrations of DOX (0.02, 0.04 and 0.2 μM; Figure 2Aii,Bii). The DOX alone treatment required a range of 50–500-fold more drug, depending on the dosage comparisons, than the combination therapies to reach sub-IC_50_ thresholds in cHCI-10 cells at the same time. Similar trends were observed at the 48-h time point (Appendix A). Taken together, these results suggest that WNT-therapy can sensitize the TNBC PDX cells to doxorubicin, but not to cisplatin. We have previously published the IC_30_, IC_50_ and IC_70_ values and plotted isobole curves to determine the combination index; the combination indexes for doxorubicin and ICG-001 were less than 1, confirming synergistic effects [10,15].

Next, we conducted assays using the IncuCyte^®^ Cytotox Green Reagent that quantifies cell death in real time. We used sub-IC_50_ concentrations of ICG-001, either at 5 µM, for MDA-MB-231 cells, or at 10 µM for cHCI-10 cells in combination with two CIS dosages at 0.5 µM or 20 µM in the MDA-MB-231 (Figure 2Ci) and 1 µM or 20 µM for the cHCI-10 cells for 48 h (Figure 2Di). Similarly, we used two DOX dosages 0.5 µM or 5 µM in the MDA-MB-231 cells (Figure 2Cii) and in the cHCI-10 cells (Figure 2Dii). The results show that the amplitude of the cytotoxicity effect of the combinatorial treatment is highest when cells are treated with DOX plus ICG-001 in both cell lines. In contrast, CIS plus ICG-001 was not effective in the MDA-MB-231 cells, but was in the cHCI-10 cells. It is known that CIS can elicit chemoresistance through hyper activity of PARP-1 in non-small cell lung carcinomas [16] and that PARP-1 regulates EMT through SNAIL expression in doxorubicin-resistant MDA-MB-231 cells [17]. We posited that PARP expression could account for the cell death difference between the two models above; thus, we immunoblotted for total PARP-1 and cleaved PARP-1 (Figure 2F). Results indicate that both DOX and CIS induce PARP-1 protein expression in MDA-MB-231 cells as expected, consistent with the literature. In contrast, this is not observed in cHCI-10 cells. More importantly, ICG-001 therapy sensitizes with both CIS and DOX to generate the cleaved PARP-1 product, but not in the MDA-MB-231 cells. BAX serves as a cell death marker control and both ACTIN and TUBULIN serve as loading controls. These data support the argument that DOX plus ICG-001 synergize to inhibit proliferation and to activate cell death by inhibition of PARP-1 activity.

### 2.3. WNT Inhibition Interferes with Simultaneous Multi-Organ Metastases in Vivo in MDA-MB-231 Cells

Over a decade ago, the Massague lab characterized MDA-MB-231 metastatic models generated by intracardiac (IC) xenografting, demonstrating that these same cells would show enhanced metastasis to the lung and bone after simultaneous repeated IC injections [18]. More importantly, they demonstrated that WNT signaling was critical for maintenance of the aforementioned metastatic enrichment models [19]. We questioned if we could: (1) Improve metastatic modeling by surgically transplanting MDA-MB-231 cells directly into the mammary fat of immunocompromised NOD/SCID Gamma (NSG) mice. (2) Generate de novo simultaneous multi-organ metastases, including lymph node and bone metastases, and (3) address one of the overarching challenges of MBC and/or metTNBC, which is the inability to treat widespread metastases, including to visceral organs (liver and lungs).

We first generated MDA-MB-231-Luc cells from parental MDA-MB-231 (stably transfected with a lentiviral luciferase vector) and bilaterally surgically transplanted ~1.25 × 10^6^ cells into the mammary fat pad of NSG mice (Figure 3). One week after surgery, we began treatment with ICG-001 via intraperitoneal injections every other day for two weeks (200 mg/kg, IP; n = 10 mice/group). This dosage of ICG-001 is the human equivalent dosage (HED) of 600 mg/m^2^, which is currently used in a phase II clinic trials (https://clinicaltrials.gov/), ICG-001/PRI-724; Figure 3Ai). We determined that relative to vehicle (Veh) treatment, ICG-001 significantly decreased tumor volume (mm^3^) as early as day 22 post-transplantation (*** *p* = 0.001), and a reduced tumor volume is maintained through the study endpoint (day 30 post-transplantation). At tumor harvest, we obtained ex vivo images of control and ICG-001-treated tumors, showing reduced overall size and angiogenesis in the ICG-001-treated tumors (Figure 3Aii; n = 3). We also longitudinally compared tumor bioluminescence flux for all the primary tumors throughout the study, since bioluminescence activity requires ATP from live cells, confirming that tumors treated with ICG-001 had reduced viability (Figure 3Bi; *p* = 0.048). These observations were verified by ex vivo bioluminescence images of mice treated with ICG-001 therapy relative to the vehicle controls. Additionally, the average endpoint tumor wet weight was significantly reduced in the ICG-001 cohort, whereas there were no significant changes in the mean animal body weights in response to therapy (Appendix A).

Untreated mice were removed from the study due to either primary tumor burden, metastatic burden and/or or deterioration of health status (the mouse was then scored as dead). Based on these criteria for survival, the Kaplan–Meier curves show that ICG-001 significantly increased survival rates (80%) (Figure 3C; *p* = 0.0092). Mechanistically, in these tumors, WNT10B/β-CATENIN direct target genes [9,10,20] (AXIN2, HMGA2, CCND1, PCNA, and MYC; Figure 3Di) and EMT markers (Vimentin and SNAIL; Figure 3Dii) showed decreased protein expression after ICG-001 therapy. Quantification of the immunoblots by ImageJ, from biological triplicate tumors, revealed that ICG-001 reduced protein levels of the aforementioned genes (Appendix A). Overall, these in vivo results are congruent with the in vitro results previously shown in Figure 1 and Appendix A.

To determine if ICG-001 could inhibit de novo simultaneous multi-organ metastases (lymph node, bone, and brain metastases) observed in our luciferase labeled MDA-MB-231 model, the frequency of metastasis to various organs was compared by conducting ex vivo bioluminescence analysis (Appendix A). In untreated cohorts, the lymph nodes, lungs and bones had a metastasis rate of 100% and other organs varied from 80% to 20% frequency. As representative images, we show two whole-body bioluminescence images from two mice each from the vehicle and treated groups (Figure 3E). ICG-001 therapy significantly decreased whole-body bioluminescence (Figure 3F) and the number of mice with metastasis (Appendix A). Moreover, bioluminescence images of ex vivo organs are shown pre- and post-therapy (Figure 3G and Appendix A). ICG-001 significantly repressed lung, liver, ovary and kidney metastasis (** *p* = 0.001; Appendix A).

Taken together, these results show that our model generates de novo simultaneous lymph node, visceral organ, bone and brain metastases following orthotopic transplantation of the labeled MDA-MB-231 cells into the mammary fat pad of NSG mice. More importantly, WNT inhibition mediated by ICG-001 monotherapy diminished or blocked metastasis to multiple organs, including the visceral organs typical of TNBC patients (lung and liver).

### 2.4. WNT Inhibition Interferes with De Novo Multi-Organ Metastases in a Chemo-Resistant TNBC PDX Model

To determine the frequency of de novo simultaneous multi-organ metastases from our luciferase-labeled TNBC chemo-resistant PDX tumor model (HCI-10Luc2) [10], we surgically implanted tumor fragments (~2 mm from a tumor-bearing donor female) into each inguinal mammary gland of 25 NSG mice. Eight weeks after transplantation, bioluminescence imaging was initiated to monitor primary tumor growth with accompanying metastases, which first developed in the axillary lymph nodes (LN) and the lungs, as previously reported Figure 4A [10]. We show two representative mice that exhibited metastases affecting multiple organs (Figure 4B). Ex vivo bioluminescence confirmed metastases in the lung, liver, ovaries, kidneys, bones and brain of these mice (Figure 4C). The frequency for de novo simultaneous metastases is as follows: lung and LN (100%), ovaries (40%), liver, spleen and kidneys (20%), bone (15%) and brain at 5% (Appendix A). More importantly, following ICG-001 therapy at 200 mg/kg (IP, every other day for two weeks), the metastatic flux measured ex vivo was reduced in most of the organs. ICG-001 also reduced primary tumor growth in a statistically significant manner, as shown by: (1) whole tumor images (Bi) (2) bioluminescence imaging of tumors (Bii) and (3) H&E staining (Biii) (Appendix A). Tumor growth reduction was observed whether monitored by calipers (Ci), by luciferase flux (Cii) or by tumor wet weight and two images of mice with measurable luciferase flux are shown (Appendix A; ** p* < 0.05, ** *p* = 0.001 and Appendix A).

Next, we compared whole-body metastases in vehicle control mice vs. ICG-001-treated mice dosed at either 100 mg/kg or 200 mg/kg (n = 10/cohort). The higher dosage resulted in statistically significant repression of luciferase flux units (Figure 4D; ** *p* = 0.0087). Mechanistically, the reduction in metastases is associated with loss of protein expression of both WNT10B/β-catenin direct target genes (AXIN2, MYC and PCNA; Figure 4Ei) and EMT markers (Vimentin and SNAI; Figure 4Eii) in the primary tumors. Quantification of immunoblots was conducted with ImageJ as previously shown (Appendix A). H&E staining of primary HCI-10Luc2 tumors and the simultaneously arising multi-organ metastases in the lungs, ovaries and fallopian tube (FT), liver and kidneys are shown for both the vehicle controls and treated groups (Figure 4F; 200 mg/kg). Loss of tumor masses in the various organs was confirmed by a reduced immunohistochemistry (IHC) signal for an anti-human mitochondrion (Hu-Mito) antibody, in the aforementioned organs.

Next, we posited that ICG-001 targets non-tumor cell mechanisms are also required for tumor growth and metastasis, such as in endothelial cells. To this end, we conducted immunohistochemistry (IHC) for CD31, a marker for endothelial cells (Figure 4G). The results showed that CD31 expression was relatively unchanged in the cHCI-10 model. In contrast, CD31 expression was upregulated in the MDA-MB-231 model in response to WNT monotherapy. Based on the above results, we used VEGFA as a marker for angiogenesis to determine if similar results would be observed, as those of CD31. VEGFA protein expression was upregulated in both models after WNT monotherapy. These results suggest that ICG-001 treatment is promoting angiogenesis during tumor outgrowth, perhaps normalizing the tumor vasculature, leading to better tumor perfusion with systemic targeting agents [21].

The above results present strong evidence that in a highly chemo-resistant TNBC PDX model, transplantation to the mammary fat pad of NSG mice can elicit de novo simultaneous multi-organ metastases. Moreover, for the first time, we demonstrate that ICG-001 monotherapy has the capacity to prevent de novo simultaneous multi-organ metastases arising in this doxorubicin-resistant TNBC PDX model. These data suggest that ICG-001 may be appropriate as a second-line therapy in TNBC patients who progress on anthracyclines.

### 2.5. Micro-Computed Tomography (μCT) Analysis of Bone Metastasis Reveals Differential Osteoclastic Properties in the MDA-MB-231 and TNBC PDX Tumors

Bone metastasis is the most common site of metastasis in women diagnosed with breast cancer, although it is less common in TNBC compared to ER+ breast cancer [4]. As determined by ex vivo bioluminescence imaging, MDA-MB-231Luc cells surgically transplanted into the mammary fat pad of NSG mice metastasized with 100% frequency to at least one leg bone, and >90% of the time to both legs Figure 5C.

To characterize the effects of metastasis on bone structure, we conducted micro-computed tomography (μCT) analysis of the bones. First, we demonstrated the baseline μCT for the NSG control female mice Figure 5A. Both femoral and tibial trabeculae are presented. The μCT of bone metastases derived from MDA-MB-231Luc cells revealed osteolytic properties in the tibial trabeculae, but not the femoral trabeculae (Figure 5B; n = 2 mice). The red arrows highlight the pores that are generated in the bone by the bone metastasis; the insert shows the corresponding ex vivo bioluminescence from those same bones. Moreover, one of the tibial trabeculae μCT images from the MDA-MB-231 model was pseudo-colored to further highlight the bone resorption. More importantly, ICG-001 therapy prevented bilateral bone metastases arising from the MDA-MB-231Luc cells (Figure 5Ci,ii; * *p* = 0.0225; n = 5 mice). Taken together, these results suggest that bone metastases arising from MDA-MB-231Luc cells respond to WNT monotherapy, as previously observed for lymph node and visceral organ metastases.

### 2.6. Combination of the WNT Inhibitor ICG-001 and Doxorubicin Blocks Simultaneous Multi-Organ and Bone Metastases in A Chemo-Resistant PDX Model

We next questioned if ICG-001 therapy used in combination with doxorubicin would simultaneously prevent multi-organ metastases and bone metastases originating from the chemo-resistant TNBC PDX tumors. To more efficiently model metastasis, we hypothesized that tail vein injections may increase the experimental-induced metastatic frequency to the bone and other organs (e.g., liver and ovaries) relative to when cells are orthotopically placed into the mammary gland. Therefore, in a pilot experiment, we conducted tail vein injections to experimentally induce metastasis using freshly disassociated HCI-10 Luc tumor cells from multiple pooled HCI-10Luc2 PDX mammary tumors. Cells were digested into small organoids, briefly cultured (2 passages) in M87 media in ultralow adhesion vessels, and then trypsinized and DNase I-treated to obtain a single cell suspension to inject via the tail vein of NSG female mice (1 × 10^6^ cells/mouse) (Appendix A n = 10 mice). The mice were tracked in vivo for bioluminescence signal longitudinally for ~6–8 weeks and then all organs were harvested and bio-imaged ex vivo. Using this approach, we observed by ex vivo bioluminescence imaging that ~80% of the bones had detectable metastasis. Additionally, we observed an increased frequency of metastasis to the liver and ovaries, amongst other organs and, as expected, 100% of lungs had detectable metastasis (Appendix A). These data provide evidence that tail vein injections with cHCI-10Luc cells increase bone metastasis frequency.

Next, we repeated the cHCI-10Luc2 tumor cell isolation process and tail vein injection for the purposes of testing combinatorial therapy with ICG-001 and DOX in vivo (Figure 6). To limit the potential toxicity of combinatorial therapy, beginning one day after tail vein injection, we treated mice with 50 mg/kg ICG-001 (IP, every other day) and with 1.4 mg/kg of DOX (IP, once every two weeks) for up to three cycles of combination therapy. At these dosages, we have previously shown that the photon flux derived from lung metastases was repressed [10]. We observed a decrease in total body bioluminescence photon flux in the ICG-001 + DOX cohort relative to the DOX-alone group (Appendix A; n = 5 per group). Panels of the bioluminescence images are shown from mice in the vehicle, DOX-alone and combination therapy groups.

Next, we conducted ex vivo bioluminescence of the liver, bone and ovaries to quantitate photon flux at the study endpoint (Figure 6A–C). As shown in the representative images from each organ for the above three cohorts, with DOX alone, there was no significant difference in the liver, bone or ovary’s metastatic signal (Figure 6A–C) relative to the vehicle control. Moreover, liver metastasis for the DOX-alone cohort had statistically higher flux-units versus vehicle-treated mice (Figure 6A.) In contrast, the DOX + ICG-001 combinatorial therapy significantly repressed liver flux compared to DOX-alone, but combination therapy did not significantly repress metastatic flux in the ovaries or bone relative to DOX alone (Figure 6B,C). These results suggest that liver, responds to the combinatorial therapy, as was the case with the lungs [10]. Similarly, the kidneys showed repression of metastases with the combinatorial therapies with lungs serving as the positive control (Appendix A).

Taken together, these results demonstrate that the addition of a WNT inhibitor to anthracycline therapy can inhibit simultaneous multi-organ metastases, including bone metastases, in a chemo-resistant TNBC PDX model.

## 3. Discussion

There is significant heterogeneity within the five breast cancer subtypes identified originally from microarray analysis [22], resulting in major therapeutic challenges. MBC is an incurable disease using currently available therapies [4,23]. TNBC patients diagnosed with both multi-organ metastases and bone metastasis have worse survival rates (less than 1 year) than at either site alone.

We have shown that the WNT10B/β-catenin/HMGA2/EZH2 signaling axis is critically important in chemo-resistant TNBC and that targeting this network can prolong survival by repressing primary tumor growth and multi-organ metastases to both the lungs and lymph nodes [10]. More importantly, the WNT10B network model for chemo-resistant TNBC serves as a relevant pre-clinical translational model to develop precise targeting therapeutics to combat metTNBC. We provide evidence that the WNT10B/β-catenin/HMGA2/EZH2 signaling axis is active in a variety of TNBC cell lines, with similar profiles to those of MDA-MB-231 cells, which have been extensively used for preclinical models for testing therapeutic efficacy [24].

Therapeutically, in vitro work with ICG-001 revealed decreased cell proliferation and decreased expression of WNT direct targets’ gene expression in a panel of TNBC models, which included two TNBC PDX-derived cell lines (treatment-naïve cHCI-2 and highly chemo-resistant cHCI-10). We quantified protein changes in all of the TNBC cell line models exposed to ICG-001 and determined that the response of the network to ICG-001 was statistically significant (Appendix A). Mechanistically, the loss of WNT/β direct targets in TNBC cells line models is in part due to the loss of EZH2-HMGA2 protein–protein interactions to complex with nuclear WNT components β-CATENIN/LEF1/TCF4 to activate WNT-targets, as we have previously shown [10].

We observed that the holistic decreases in the protein expression of these biomarkers for all the TNBC models was inversely linked to that cell line’s calculated IC_50_ value. For example, the required IC_50_ concentration for the chemo-naïve TNBC PDX cHCI-2 model (7.7 µM ± 2.5) correlated with the lowest *p = value* (0.0005) responding to ICG-001 therapy. Inversely, the highest IC_50_ concentration value for the highly doxorubicin chemoresistant TNBC PDX cHI-10 cells (31.6 µM ± 5.6) had the lowest *p = value* (0.012). These observations suggest that each cell line has a different oncogenic nuclear β-CATENIN activity, with higher oncogenic activity requiring greater dosages of drug exposure to block β-CATENIN activity.

To support the concept that ICG-001 may be best paired with cytotoxic drugs, given the TNBC subtype molecular heterogeneity, we found that ICG-001 sensitized TNBC cells to doxorubicin in both MDA-MB-231 and cHCI-10 cells when compared to cisplatin. Since both cisplatin and doxorubicin have been linked to PARP-1-associated chemoresistance in MDA-MB-231 cells [16,17] we provide evidence that cleaved PARP-1 occurred only in the TNBC cHCI-10 model. These results suggest that WNT inhibition maybe useful in combination with current PARP inhibitors to target not only TNBC, but those patients who are carriers of BRCA1 mutations. In support of this notion, we have previously shown that our WNT10B-network consisting of WNT10B/β-CATENIN/HMGA2/EZH2, which is present in early lesions of BRCA mutation carrier women, as well as the primary tumors and metastatic lesions. To expand these findings further, ICG-001 will need to be evaluated with other National Comprehensive Cancer Network (NCCN) recommended therapies for metTNBC, such as Olaparib, amongst others.

One of the goals of this research was to improve preexisting in vivo metastatic models that rely on the technically challenging intra-cardiac injection [18,19,23]. In addition, we set out to determine whether ICG-001 has efficacy against de novo simultaneous multi-organ metastases originating from the mammary fat pad, which must complete the entire metastatic cascade, unlike IC injection, with and without associated bone metastases in the same animal; the goal was to evaluate treatment in an animal model with extensive stage IV disease. We provide evidence that MDA-MB-231 cells originating from the mammary fat pad of NSG mice metastasize to both the lungs and to the axillary lymph nodes with identical frequencies, but that the frequencies to other organs are different in each animal. Secondly, we provide strong evidence that the aggressive MDA-MB-231 model responded to ICG-001 monotherapy at 200 mg/kg but not at 50 mg/kg [25], as metastases were inhibited in the lungs, lymph nodes, livers, ovaries and bones. We observed this also with cHCI-10Luc2 cells [25]. Based on the fact that cancer metastasis requires that primary tumors cells evolve the capacity to intravasate into lymphatic system or the vasculature to metastasis to distant organs, we posited if WNT monotherapy could affect other cells than the highly proliferating tumor cells? Could response in vivo to WNT monotherapy also effect endothelium cells and/or angiogenic markers? To this end, we showed that the endothelial marker CD31 was upregulated only in the MDA-MB-231 primary tumors, but was not in cHCI-10 PDX primary tumors. In contrast, the angiogenic marker VEGFA was upregulated in both models after ICG-001 therapies. These differential results suggest the differences in vascular mimicry and/or tumor-cell-intrinsic angiogenic response may be an important driver of metastasis in some tumors. It remains to be tested if ICG-001 normalizes the tumor-associated vasculature to permit better systemic therapy infusion into the tumor or if WNT monotherapy may cooperate with antiangiogenic therapies.

Bone metastasis is the most common site for breast cancer patients [26]. Both the WNT signaling gene signature [5] and WNT10B network are associated with TNBC metastasis [10]. TNBC is more frequently associated with lung and brain metastasis, with less frequency to the bone [3]. Bone metastases originating from MDA-MB-231 primary tumors were four times more frequently observed (80%) than for cHCI-10 cells (20% frequency). Micro-computed tomography (μCT) analysis demonstrated that osteoclastic activity was restricted to the tibial trabeculae, but not femoral trabeculae in the MDA-MB-231 cells. More importantly, we were able to treat bone metastasis with WNT monotherapy. Future studies with ICG-001/PRI724 + DOX could be tested with either bisphosphonate or denosumab therapy, which mechanistically targets osteoclast activity, preventing bone loss, to determine if the prevention of osteoclastic activity would significantly increase survival in either the HCI-10 PDX or the MDA-MB-231 models.

Treatment of the doxorubicin-resistant TNBC PDX model with the WNT inhibitor (ICG-001/PRI724) and the anthracycline-based treatment (DOX) blocked lung metastasis [10]. Mechanistically, we demonstrated that ICG-001 + DOX have synergistic interactions, by increasing the BCL2/BAX ratio [27], thereby enhancing apoptosis. Outside of patients who are BRCA1/2 mutation carriers, who can receive PARP inhibitors, there are no specific, targeted therapeutics for TNBC other than general cytotoxic chemotherapies. NCCN provides guidelines that recommend treating metTNBC with single agent or combination chemotherapeutics, including doxorubicin, as was the case for the patient from whom the HCI-10 PDX tumor was established [28]. It should be noted that the dose of DOX used in our present study (1.4 mg/kg) is only the human equivalent dosage (HED) of 45.96 mg/m^2^, which is approximately 25–40% less than the NCCN guidelines recommend if doxorubicin is used at as a single agent. This suggests that we may be able to decrease the DOX dose when used with the WNT inhibitor, with a resultant decrease in toxicity [29]. Here, we provide further evidence that ICG-001 can synergize with DOX in vivo to decrease liver and bone metastases, as well as the metastases found in other organs that are less commonly observed in TNBC (kidney). Interestingly, the ovaries did not significantly differ in the ICG-001 + DOX combination therapy relative to the either vehicle or DOX-alone groups, suggesting a potentially unique microenvironment of the ovaries that interferes with response to therapy. In support of this hypothesis, the patient from whom the cHCI-10 PDX tumor was generated had ovarian metastasis that was unresponsive to anthracycline-based therapy [28].

## 4. Materials and Methods 

**Chemicals**: ICG-001 (#A8217) was purchased from ApexBio, Doxorubicin was purchased from Sagent Pharmaceuticals and Cisplatin was purchased from Sigma-Aldrich (St. Louis, MOS, USA).

**Drug dosages for IC_50_, proliferation, cytotoxicity assays, qPCR and immunoblotting**: Calculation of IC_50s_ and proliferation WST-1 assays with ICG-001 alone used log-range dosages from 0.02 µM to 30 µM. Cisplatin alone ranged from 0.02 µM to 20 µM and doxorubicin alone ranged from 0.02 µM to 30 µM. The combinatorial therapies with ICG-001 plus cisplatin and/or doxorubicin are described in the figures and figure legends for the WST-1 assays and the cytotoxicity assays using IncuCyte^®^ Cytox-Green reagent (Ann Arbor, MI, USA). All of the experiments for qPCR and/or immunoblotting were conducted with each cell line model’s calculated IC_50_ value as shown in Appendix A. 

**Cell Culture Assays**: All conventional cell lines were maintained in a humidified atmosphere with 5% CO_2_ in DMEM plus 1% Pen/Strep and 10% FCS (Omega Scientific, Norwalk, CT, USA). All cells were purchased from ATCC, and authenticated by Genetica prior to use in experiments. Development and culture of cHCI-10 and cHCI-2 PDX cells and all other cell lines have been previously described in detail [10]. All TNBC cells were synchronized at G1 for cell cycle progression analysis as was described elsewhere (19). For all the ICG-001 experiments, we synchronized cells as previously described [20] and the treatments began 16 h after release to S-phase for an additional 48 h. Cell proliferation was measured using WST-1 cell proliferation assay (Sigma-Aldrich, St. Louis, MOS, USA) according to the manufacturer’s protocol. All experiments were conducted with both three biological and technical triplicates and growth inhibition was measured in comparison to the corresponding vehicle-only control wells.

**Tumor Biology, Metastasis Experiments and PDX Tumor Modeling**: Transplantation and xenograft experiments were carried out using standard surgical procedures as previously described [10]. Briefly, tumor cells were surgically transplanted into the inguinal mammary fat pads of NSG mice. ICG-001 therapy was used at 100 mg/kg and/or 200 mg/kg every two days, administered by IP injections for up to two weeks. When used in combination with doxorubicin (1.4 mg/kg, IP every two weeks), the ICG-001 dose was reduced to 50 mg/kg. All mice were euthanized at nine weeks after surgical transplantation. UTHSC IACUC committee approved all of the animal studies.

**Micro-computed tomography (μCT) analysis**: Tibias and femors from 12–14-week-old NSG female mice were stored in PBS and then scanned using a Scanco μCT 40 (Brüttisellen, Switzerland) set at 55 kVp/109 µA. The entire femur and tibia were scanned in the sample holder with 12.3 mm diameter at medium resolution. These tubes were filled with PBS and the top of the tube was covered with Parafilm (American National Can, Chicago, IL, USA) to prevent dehydration. A scout view of each bone was taken and the sample height was adjusted to ensure the bone was within the field of view. The tibias and femora images were obtained at 6 μm resolution. The integration time and Gaussian filter used for these samples was 300 ms and 1, respectively. Solid three-dimensional models were reconstructed from these images automatically after completion of each cone-beam image stack with the built-in software. The trabecular parameters were calculated on 200 slices of trabecular bone from a region just below the growth plate as described in Khalid and colleagues [30,31].

**Statistical methods**: For all the experiments, statistical analysis was performed using Prism 5 software including Student’s *t*-test and one-way ANOVA, and all graphs were generated using Prism 5 software. Unless otherwise stated, all error bars were calculated and represented in terms of the mean ± SEM.

**Ethics approval and consent to participate**: The PDX models used for this work were provided by Alana Welm (Huntsman Cancer Institute, Salt Lake City, UT, USA), who provided flash frozen fragments of parental (non-labeled) HCI PDX models for downstream Western blotting, and cryopreserved tissue fragments to passage the PDX models in NSG female mice at UTHSC.

**Ethics approval**: UTHSC IRB and IACUC committees have approved all of the human (#15-04009-XP and #14-03564-XM) and animal work (#19-014 and # 18-121_ conducted for this manuscript.

## 5. Conclusions

We provide pre-clinical evidence in vitro and in vivo that WNT inhibition is effective as a single agent, and when used in combination with a low dose of doxorubicin in TNBC that this combinatorial regimen can be potentially translated into clinical trials, providing a novel therapeutic approach for patients with this deadly cancer.

## Figures and Tables

**Figure 1 cancers-11-02039-f001:**
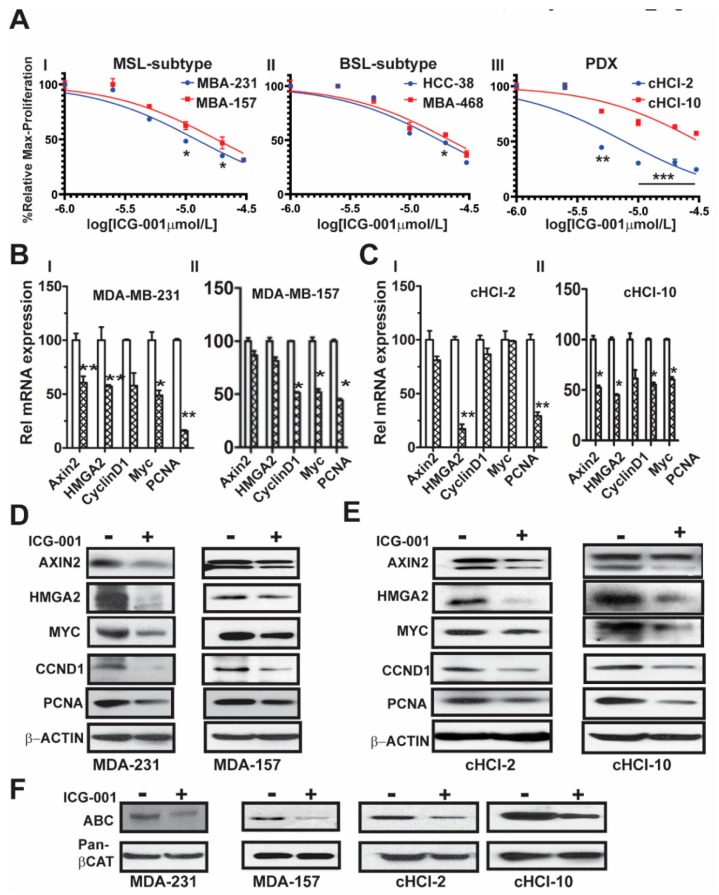
Determination of differential IC_50_ (DIC_50_) of ICG-001 on multiple TNBC cell lines. (**A**) Results of WST-1 proliferation assays following 48 h of treatment with ICG-001 at various dosages ranging from 0.2 μM–30 μM using MDA-MB-231, MDA-MB-157, HCC38, MDA-MB-468 and TNBC PDX patient-derived cHCI-2 and cHCI-10 cells. All experiments in panels (**B**–**E**) used the cells’ specific IC50s as shown in Appendix A. (**B**,**C**) qPCR for *AXIN2, HMGA2,*
*CCND1**, MYC* and *PCNA* in some of the same cells from panel A. (**D**,**E**) Immunoblot analysis for AXIN2, HMGA2, MYC, CCND1 and PCNA. (**F**) Immunoblot for non-phosphorylated Active-β-CATENIN (ABC) and total-β-CATENIN and are shown. β-ACTIN serves as the loading control. Results are expressed as mean ± SE, n = 3; unpaired Student’s *t*-test, two-tailed unequal variance used to calculate *p* values; *** *p* = 0.001, ** *p ≤* 0.01 and * *p* ≤ 0.05 vs. control. Details of western blot can be viewed at the Appendix A.

**Figure 2 cancers-11-02039-f002:**
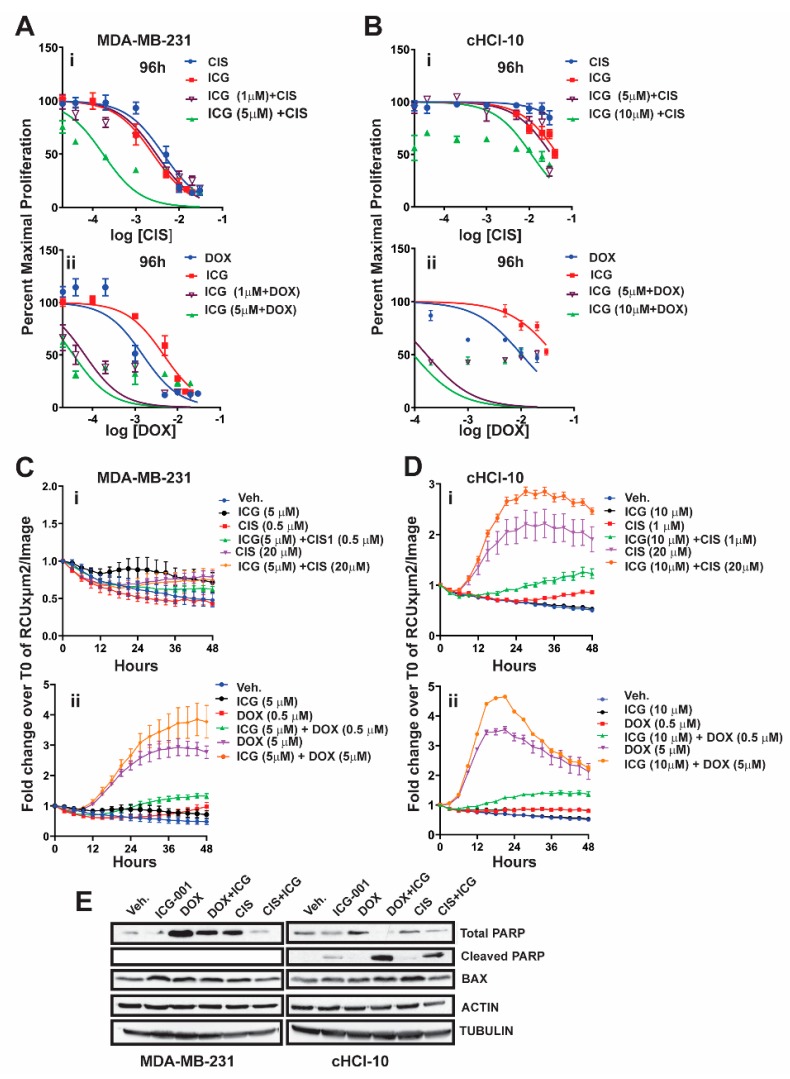
ICG-001 is able to synergize with doxorubicin, but not cisplatin, to repress tumor cell proliferation and to increase cytotoxicity in the doxorubicin chemoresistant HCI-10 PDX TNBC cells. MDA-MB-231Luc (**Ai**,**ii**) and cHCI-10Luc (**Bi**,**ii**) cells were analyzed by WST-1 assays at 96 h, following exposure to ICG-001 (1 µM and 5 µM concentration, MDA-MB-231 or 5 µM or 10 µM concentrations for cHCI-10 cells) in combination with cisplatin or DOX at various increasing concentrations, demonstrating inhibition of tumor cell proliferation. IncuCyte^®^ Cytox Green reagent was used to measure cytotoxicity in MDA-MB-231 (**Ci**,**ii**) and cHCI-10 (**Di**,**ii**) cells at various combinatorial concentrations over 48 h, showing an increased cytotoxicity response with ICG plus DOX. (**E**) MDA-MB-231 and cHCI-10 cells were exposed to ICG-001 (MDA−MB−231 = 5 µM; cHCI-10 at 10 µM) alone or in combination with DOX (0.5 µM) or with cisplatin (0.5 µM) for 48 h. Immunoblotting for total PARP, cleaved PARP, and BAX was conducted. ACTIN and TUBULIN served as loading controls. Results are expressed as mean ± SEM, n = 3. Details of western blot can be viewed at the Appendix A.

**Figure 3 cancers-11-02039-f003:**
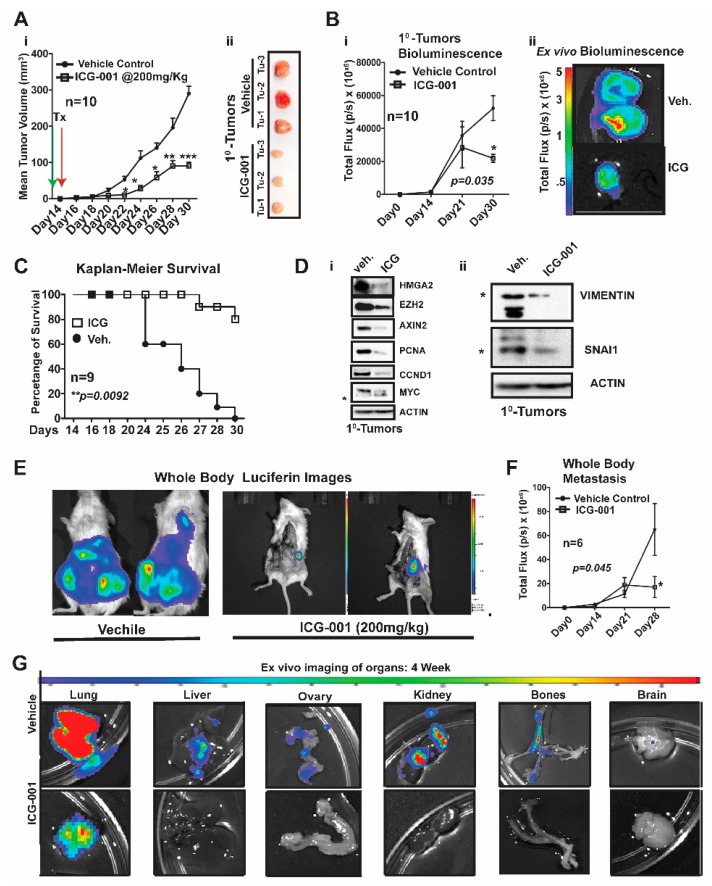
WNT inhibition interferes with simultaneous multi-organ and bone metastases in vivo in MDA-MB-231 cells. MDA-MB-231 stably transduced with lentivector-luciferase was used to track cells by bio-imaging after surgical transplantation into the mammary fat pad of NSG mice, beginning one week after initiation of ICG-001 therapy (200 mg/kg, IP every other day for two weeks). (**Ai**) Tumor volume was tracked over time using calipers (n = 10 mice). (**Aii**) Representative images of tumors from the vehicle and ICG-001-treated mice. (**Bi**) Total bioluminescence flux (photons/sec, p/s) was quantified longitudinally in the primary tumors. Standard deviation is shown. (**Bii**) Ex vivo bioluminescence images of tumors. (**C**) Kaplan–Meier Survival curve from the vehicle and ICG-001-treated mice, n = 9/group. (**D**) Immunoblot analysis of HMGA2, EZH2, AXIN2, PCNA, CCND1 and MYC (**i**) and EMT markers VIMENTIN and SNAI from the primary tumors and (**ii**), β-ACTIN serves as the loading control. (**E**) Representative whole-body luciferase reporter images from vehicle or ICG-001 treated mice. (**F**) Quantification of whole-body metastasis as measured by the total bioluminescence flux in (photons/sec, p/s’ n = 6 mice/group). (**G**) Ex vivo bioluminescence of the organs. *P-values* were generated by one-way ANOVA followed by pairwise Student’s *t*-tests (* *p* < 0.05, ** *p* < 0.01 and *** *p* < 0.001). Total light flux was compared in the whole-body of the mice after therapy. Details of western blot can be viewed at the Appendix A.

**Figure 4 cancers-11-02039-f004:**
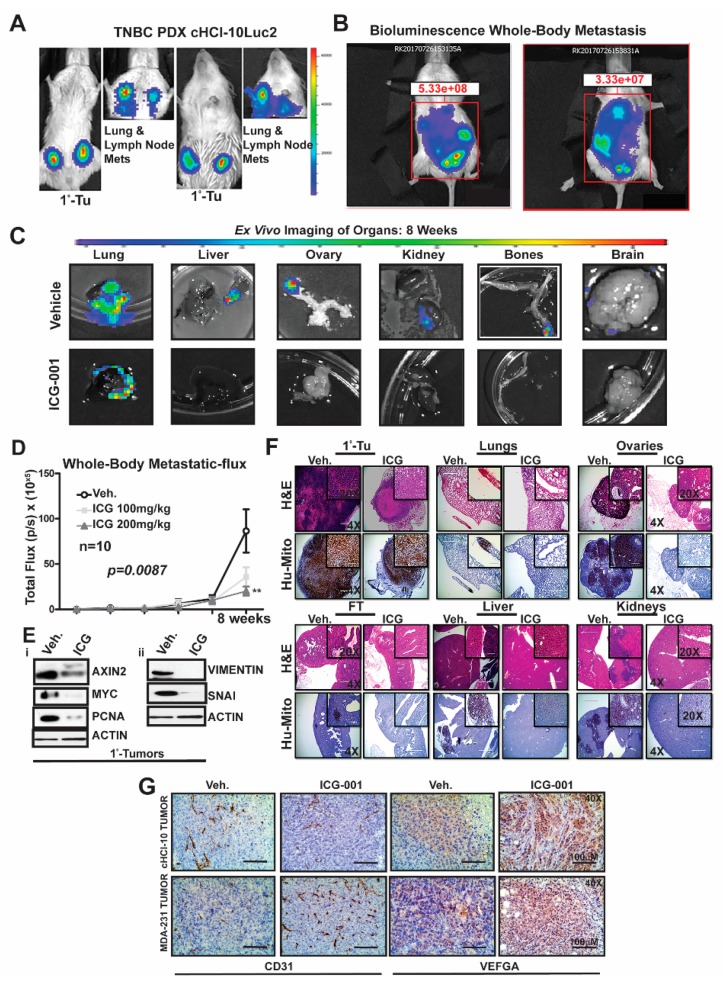
WNT inhibition interferes with de novo whole-body metastasis in highly chemoresistant TNBC PDX model. The TNBC PDX tumor model HCI-10, which is stably transduced with luciferase to track cells in vivo was bilaterally transplanted into the mammary fat pad of NSG mice. Three weeks after transplantation, ICG-001 therapy was initiated at a dose of either 100 or 200 mg/kg (IP, every other day for two weeks). (**A**) Bioluminescence images of both the primary tumors and either the lungs or lymph nodes at 6-weeks post-transplant. (**B**) Representative images of mice with whole-body metastases at 8 weeks after transplantation. (**C**) Ex vivo bioluminescence of lung, liver, ovary, kidney bone and brain harvested from the vehicle and ICG-001-treated mice. (**D**) Total light flux was compared in the whole body of the mice after therapy. (**E**) Immunoblot analysis of AXIN2, MYC (i) and VIMENTIN and SNAIL (ii) in primary tumors (ICG-001 at 200 mg/kg). β-ACTIN serves as the loading control. (**F**) Anti-metastatic effects of ICG-001 are shown, along with micrographs of H&E staining and anti-human mitochondria antibody staining of harvested tissue to confirm the presence of human tumor cells in rodent organs. (**G**) IHC was performed for CD31 and/or VEGF-A in the vehicle or ICG-001 treated cohorts for both the cHCI-10 or MDA-MB-231 primary tumors. *p*-Values were generated by one-way ANOVA followed by pairwise Student’s *t*-tests (** *p* < 0.01). Details of western blot can be viewed at the Appendix A.

**Figure 5 cancers-11-02039-f005:**
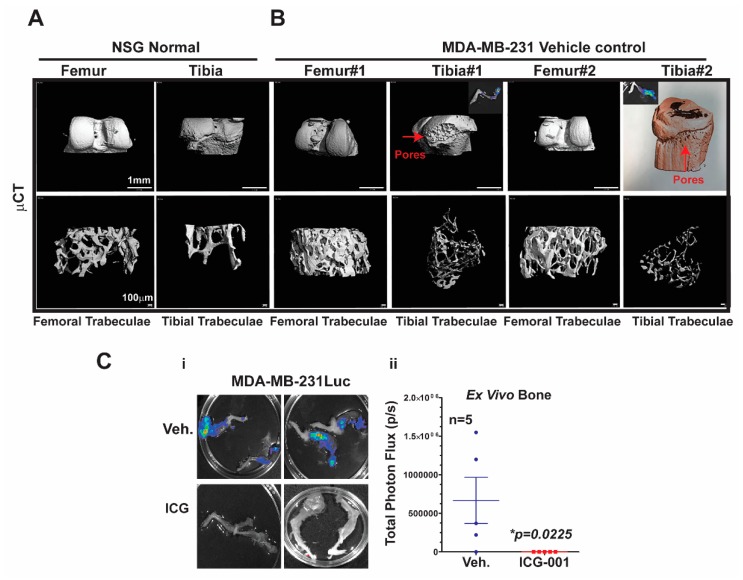
Impact of bone metastases on the bones as analyzed by high-resolution micro-computed tomography (μCT). (**A**) 3D images of a femur and a tibia from a control non-tumor-baring NSG mouse at 12 weeks of age, showing normal bone phenotype in both femoral and proximal tibial trabeculae bone. (**B**) Bones bearing metastases derived from MDA-MD-231Luc cells show the femoral and proximal tibial trabeculae; two representative mice are shown from the vehicle and ICG-001-treated mice. Red arrows (the pores) highlight loss of bone mass only in the tibial trabeculae. (**C**) Representative ex vivo bioluminescence images of bones from the vehicle and ICG-001-treated mice (i) and the quantification of flux-units (ii; n = 5 mice) at study endpoint. Inserts show the ex vivo bioluminescence images obtained for the same bones that were analyzed by μCT. *p*-Values generated by one-way ANOVA followed by pairwise Student’s *t*-tests (* *p* = 0.0225). Scale Bar of 1 mm or 100 μm (A,B).

**Figure 6 cancers-11-02039-f006:**
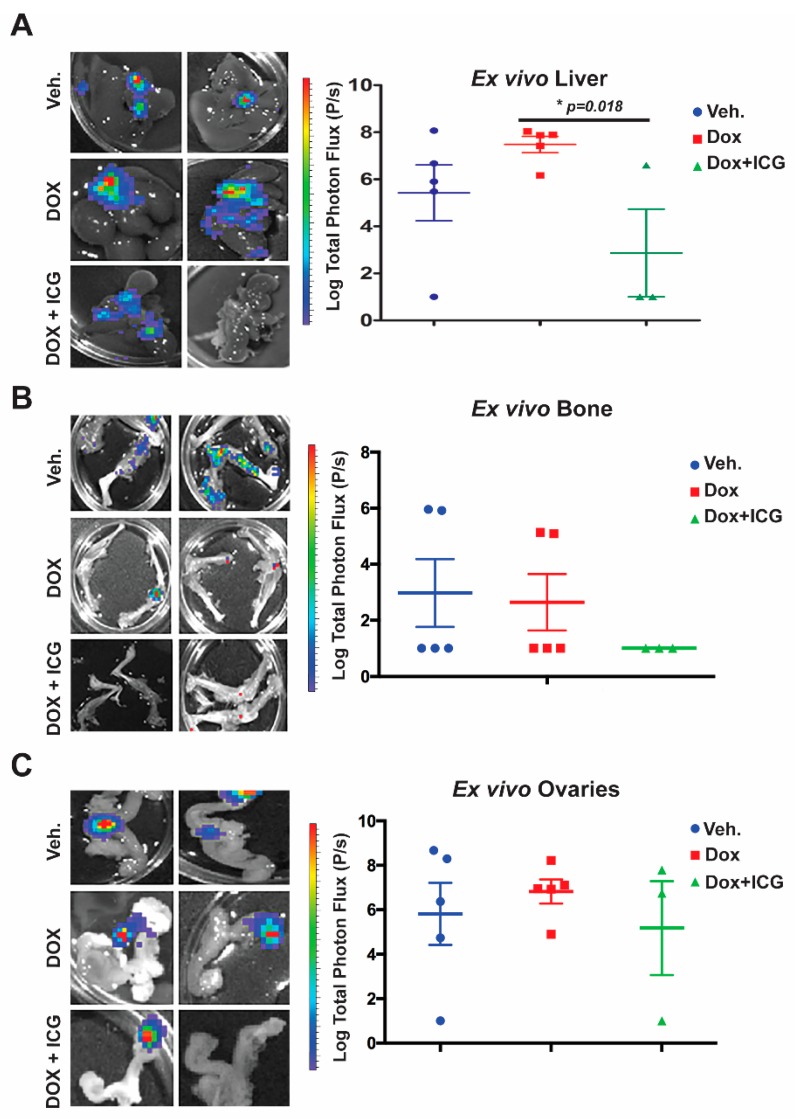
ICG-001 sensitizes chemoresistant TNBC PDX tumor cells to doxorubicin, preventing liver, bone and ovarian metastasis. cHCI-10 cells (1.25 × 10^6^) that were freshly isolated from primary PDX HCI-10Luc2 tumors were tail vein injected into NSG females. One day after tail vein injection, mice were treated with either DOX alone (1.4 mg/kg, IP) or DOX in combination with ICG-001 (50 mg/kg, IP) using the dosing schedules outlined in the materials and methods. Total flux (p/s) was quantified by ex vivo bioluminescence imaging of the liver (**A**), bone (**B**) and ovaries (**C**); *p*-Values generated by unpaired Student’s *t*-test; two-tailed unequal variance (* *p* < 0.018).

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
