# Peer review of "Simultaneous Multi-Organ Metastases from Chemo-Resistant Triple-Negative Breast Cancer Are Prevented by Interfering with WNT-Signaling"

_cancers, 2019, doi:10.3390/cancers11122039_

Round 1

Reviewer 1 Report

The author has addressed for all my concern. However, there are still minor concern to be adressed.
Figure SF1: This figure shows IC50 of cell lines. Why didn't the author show higher concentration to be reached less than 50% for survival fraction. How did the author evaluate IC50? "Beast Cancer Cell Lines" of graph in Figure SF1 does not fit as the title, because this figure contains both data from cancer (MCF-7 and SKBR3) and non-cancer cell lines (HUMEC and MCF-10A). This graph compare cancer cell lines with non-cancer cell lines. Here, in tems of "non-cancer" doesn't mean actual "control": i,g., this is different category of dose zero in dthe dose dependency or beta-actin mRNA level for RT-PCR. I suggest you to delete "Beast Cancer Cell Lines" in the Panel A to shows simply "Cell lines" just above name of cell lines in the right of Panel A.
According to comments on ATCC, "the MCF 10A cell line is a non-tumorigenic epithelial cell line". What does mean "normal-TNBC" for MCF-10A? The author states that definition of TNBC is triple negative breast cancer.

Author Response

Point by point response:

We have gone back to our data and have added one more concretion point to the HUMEC cell line only and made the correction to SF1A, as requested. We have also amended SF1B to clarify and changed nomenclature, as requested. We also corrected nomenclature for MCF10A and referred them as normal epithelial rather than Normal-TNBC on page 3, lines 99-100

We made the following changes to page 4 , line 168, as follows:

“We have previously published the IC30, IC50 and IC70 values and plotted isobole curves to determine the combination index; the combination indexes for doxorubicin and ICG-001 were less than 1, confirming synergistic effects10,15

This manuscript is a resubmission of an earlier submission. The following is a list of the peer review reports and author responses from that submission.

Round 1

Reviewer 1 Report

Fatina et al described that inhibition of wnt signaling prevents multi-organ metastasis.

This has serious problems, so that this is not reached criteria of Cancers in its present form.

Major concern:

1.         Proliferation data includes large variation (Figure 2, Supplementary Figure SF2). In addition, there is extremely low dose-dependency. Therefore, IC50 values with very small variation are unreliable.

2.         All western blots are needed to be showed statistical significance.

3.         Figure 1: Effect of ICG-001 on normal cells are not only referenced but also should be confirmed by own self.

4.         Figure 3: Tumor volume means what? Is it sum of primary and secondary? What was cause of death? Shows statistical significance for metastasis to lung, kidney, and brain.

5.         Figure 6: Non-parametric one-way ANOVA is followed by Kruskal-Wallis test or Friedman test but not, atleast, Student’s t-test.

6.         Figures showed effect of ICG-001 for proliferation and metastasis with only parallel data of intra-cellular signaling molecules.

Reviewer 2 Report

The study by Fatima, El Ayachi and colleagues is an extension of earlier work done by the same group investigating WNT/β-catenin inhibition in triple negative breast cancer models (El Ayachi, Fatima et al, Can Res 2018). Using xenograft models, the authors demonstrate that blocking WNT/β-catenin signaling with the small molecule inhibitor ICG-001 results in reduction of TNBC metastasis to various organs. They also show that the combination of ICG-001 with doxorubicin is more effective at reducing metastasis than doxorubicin alone. This is an interesting study, but there are several issues that should be addressed as noted in the comments below.

Specific comments:

1.     Using a WST-1 assay, the authors demonstrate that ICG-001 reduces proliferation of different TNBC cell models. Considering that proliferation was reduced to 25% at the highest concentration, I suggest complementing these data with an additional assay that can assess toxicity. Does ICG-001 act by inhibiting proliferation or inducing cell death?

2.     Even though the authors show proliferation results from the different cell lines with a range of concentrations of ICG-001, they show these as bar graphs rather than dose response curves. It would be helpful to see these data plotted as dose response curves along with the IC50 values that are shown in the supplement.

3.     For the gene expression and immunoblotting experiments, what concentration of ICG-001 was used? This information should be included in the methods section and figures or figure legends.

4.     In Fig. 2, the authors state that ICG-001 can synergize with doxorubicin in TNBC cells that are resistant to doxorubicin alone. The authors should support this conclusion by calculating a drug synergy score by combination index or similar method.

5.     Using a mammary fatpad injection model, the authors state that ICG-001 is able to reduce dissemination of tumor cells to multiple organs and the bones of the injected mice. It would be helpful to see whole body images of the control and treated mice in addition to the example ex vivo images that are presented. These could be more compelling than the H&E stained tissue sections that are presented in Fig. 3G and 4F.

6.     Fig. 5A and B are confusing. Does Fig. 5A just depict NSG that have not been injected with MDA-MB-231Luc cells, but were treated with vehicle? And for Fig. 5B, which bones are from vehicle control MDA-MB-231Luc mice and which bones are from ICG-001 treated mice? The same questions apply to Fig. 5D.

7.     In Fig. 6A, the log total photon flux for the liver is higher for the doxorubicin treated mice than the vehicle control mice, while the doxorubicin plus ICG-001 mice show a similar photon flux as the control mice. Thus, the data are not entirely convincing that doxorubicin plus ICG-001 would be an effective combination. It would be useful to see the activity of ICG-001 alone, since it’s difficult to assess with the presented results if the reduction in metastasis is due to ICG-001 alone or the combination with doxorubicin.

8.     The authors state that based on the presented results, inhibition of multi-organ and bone metastases by ICG-001 is due to the direct effect of the drug on the tumor cells. Have the authors assessed the effect of the inhibitor on angiogenesis? This should be relatively straight forward to measure by staining tumor sections with an endothelial cell marker.